# Cancer and Aging: Two Tightly Interconnected Biological Processes

**DOI:** 10.3390/cancers13061400

**Published:** 2021-03-19

**Authors:** Lieze Berben, Giuseppe Floris, Hans Wildiers, Sigrid Hatse

**Affiliations:** 1Laboratory of Experimental Oncology, Department of Oncology, KU Leuven, 3000 Leuven, Belgium; lieze.berben@kuleuven.be; 2Department of Pathology, University Hospitals Leuven, 3000 Leuven, Belgium; giuseppe.floris@uzleuven.be; 3Laboratory of Translational Cell and Tissue Research, Department of Imaging and Pathology, KU Leuven, 3000 Leuven, Belgium; 4Department of General Medical Oncology, University Hospitals Leuven, 3000 Leuven, Belgium

**Keywords:** cancer, aging, biomarkers

## Abstract

**Simple Summary:**

As life expectancy is increasing, the older population is rapidly growing. However, older patients with cancer are still underrepresented in clinical trials, making treatment of these patients challenging for oncologists. Robust biomarkers that reflect the body’s biological age can be helpful to provide older patients with cancer with an optimal personalized treatment. However, to be able to identify such biomarkers, more in-depth research is needed in this underexplored population. In this review, we have put together the current knowledge concerning the mechanistic connections between aging and cancer, as well as aging biomarkers that could be useful in the field of geriatric oncology.

**Abstract:**

Age is one of the main risk factors of cancer; several biological changes linked with the aging process can explain this. As our population is progressively aging, the proportion of older patients with cancer is increasing significantly. Due to the heterogeneity of general health and functional status amongst older persons, treatment of cancer is a major challenge in this vulnerable population. Older patients often experience more side effects of anticancer treatments. Over-treatment should be avoided to ensure an optimal quality of life. On the other hand, under-treatment due to fear of toxicity is a frequent problem and can lead to an increased risk of relapse and worse survival. There is a delicate balance between benefits of therapy and risk of toxicity. Robust biomarkers that reflect the body’s biological age may aid in outlining optimal individual treatment regimens for older patients with cancer. In particular, the impact of age on systemic immunity and the tumor immune infiltrate should be considered, given the expanding role of immunotherapy in cancer treatment. In this review, we summarize current knowledge concerning the mechanistic connections between aging and cancer, as well as aging biomarkers that could be helpful in the field of geriatric oncology.

## 1. Introduction

As life expectancy has increased dramatically over the past decades, older persons represent a rapidly growing section of our population. This results in an increasing number of older patients with cancer as well. Moreover, as cancer and aging are closely interrelated, cancer incidence is higher in the older age categories compared younger age categories [1,2]. There are several factors associated with aging that could explain this. Firstly, there is an accumulation of oxidative stress and DNA damage over the years that is caused by a life-long exposure to endogenous metabolic insults (e.g., free radicals) and exogenous factors (e.g., UV irradiation, foods, etc.). This may eventually lead to cell transformation and tumor initiation. Secondly, senescent cells accumulate during the aging process and exhibit a senescence-associated secretory phenotype (SASP); this means that they secrete inflammatory mediators (e.g., interleukin (IL)-6, IL-8, monocyte chemoattractant protein (MCP)-2, growth-regulated oncogene alpha (GROα), etc.) that may promote tumor growth by creating a tumorigenic environment [3]. Finally, a progressive decay of immune function occurs in older individuals, whereby an effective immune response against developing tumors may fail [4]. As the aging rate is unique, biological age of a person can differ from the chronological age and two people of the same calendar age can show a different biological aging profile. Consequently, one individual could tolerate more aggressive treatments than another individual of the same age could. In addition, chemotherapy by itself is also believed to accelerate the aging process, which may result in premature aging and frailty [5]. Additionally, more extensive and intensive research is needed to gain a better understanding of the impact of aging on tumor immunity. As aging has a substantial impact on the immune system, it could influence the effectiveness of the upcoming immunotherapies [6,7,8]. As for now, we do not have optimal tools available for estimating the “biological” age of patients, which is far more relevant than their chronological age.

Clinical decision-making in geriatric oncology is currently suboptimal, and many older individuals are under-treated while others are over-treated. Under-treatment may lead to increased risk of relapse and higher mortality while over-treatment can lead to a decreased quality of life due to toxicities [9,10,11]. There is thus an urgent need for better tools to select cancer patients for specific therapies. These tools should be prognostic, providing information on the patient’s life expectancy, and also predictive for the therapeutic benefit that will be achieved by the treatment. Aging biomarkers are promising candidate tools for this purpose, which definitely merit profound investigation in a clinical setting. Several candidate aging biomarkers have been described in the literature but to date, none of these have found their way from bench to bedside [12,13,14,15,16,17,18,19,20,21,22,23,24]. In this review, we discuss the interplay between aging and cancer based on the current literature, together with potential promising biomarkers that could be useful in geriatric oncology.

## 2. Mechanistic Interface between Aging and Cancer

Aging has been acknowledged as a major risk factor for developing cancer. Moreover, the two are highly interrelated and this is reflected by the numerous shared underlying mechanisms. The following three elements involved in the aging process, also play significant roles in cancer [25,26,27].

### 2.1. Cellular Damage and DNA Damage Response

Accumulation of cellular damage is probably the most important driver of both aging and cancer, and can be caused by several shared events. Life-long exposure to many endogenous (e.g., free radicals) and exogenous (e.g., UV radiation, foods, etc.) stress factors can induce an increase of oxidative stress, leading to genomic instability and ultimately DNA damage [25,26,27]. Reactive oxygen/nitrogen species can react with DNA and can cause several types of DNA damage such as oxidation of purines and pyrimidines, single strand breaks and double-strand breaks [28,29]. Antioxidant systems like antioxidant enzymes (i.e., catalase and glutathione peroxidase), vitamins (i.e., vitamins C and E) and other radical scavengers (glutathione) are able to prevent oxidative DNA damage. Yet, when this first line defense is ineffective, the DNA damage response is triggered and the cell cycle is arrested to allow DNA repair mechanisms to restore the damage (Figure 1) [28,29]. Essential nonredundant players of the DNA damage response are ataxia-telangictasia-mutated (ATM), recruited after a double strand break via the DNA damage sensor complex MRE11-RAD50-NSB1, and ataxia-telangictasia-Rad3-related (ATR), induced after a single strand break detected by RPA, ATRIP or the RAD6-RAD1-HUS1 complex. In their turn, ATM and ATR activate several DNA damage mediators, such as breast cancer gene 1 (BRCA1) and downstream kinases like checkpoint kinases (CHK) 1 and 2. The latter are able to trigger effectors like P53, ensuring initiation of either apoptosis, cell cycle arrest or DNA repair [30]. The two most important mechanisms are base excision repair and nucleotide excision repair [28,29]. When the DNA repair mechanisms are also unsuccessful, either the apoptotic pathway or senescence program can be activated (as explained in more detail below in Section 2.2) to eliminate cell carrying potentially dangerous mutations that can lead to cell transformation and tumor initiation [25,26,27,28,29].

Besides direct DNA damage, telomere shortening has been shown to trigger a DNA damage response. Telomere length decreases with aging as telomeres shorten with each cell division. Short telomeres are associated with genomic instability, which occurs in early tumor development, and are a significant risk factor for aging-related diseases [25,26,27].

### 2.2. Cellular Senescence

Cellular senescence is considered to be one of the most important driving forces of the aging process [25]. It is triggered by factors associated with cellular damage, such as oxidative stress, and telomere shortening and expression of oncogenes [31]. The triggers activate several senescence genes, which initiate the actual induction of cellular senescence. The two senescence genes that play critical roles are *TP53* and *P16^INK4a^*, both are tumor-suppressor genes. The *TP53* gene encodes for the P53 protein and is an activator of the cyclin dependent kinase (CDK) inhibitor P21, which will inhibit CDK4/6 activity. The *P16^INK4a^* gene makes part of the *CDKN2a* or *INK4a/ARF* locus, which based on alternative splicing, encodes two protein products: P14^ARF^, a regulator of P53 stability and the P16^INK4a^ protein, an inhibitor of CDK4/6. Thus, both P53 and P16^INK4a^ prohibit the activation CDK4/6, thus blocking the phosphorylation of the retinoblastoma protein (pRB). As a result, the cell cycle is arrested in the G_1_ phase [32,33]. In Figure 2 an overview of the pathways involved in cellular senescence is shown.

Senescent cells enter a state of irreversible growth arrest, yet they remain metabolically active. As a result, damaged cells (such as cells having oxidative, DNA damage, shortened telomeres, genomic instability, oncogenic mutations) are unable to proliferate in an uncontrolled manner, which is an important antitumor mechanism. A schematic overview of the antitumor activities of cellular senescence is shown in Figure 3 [33]. When a cell is damaged, several scenarios are possible. An antiproliferative response can be activated by which the cell becomes apoptotic or it can enter senescence. If this does not occur, the cell is able to continue replication and may form a lesion. At this stage again, both apoptotic and/or senescence programs can be activated. However, when this fails, the lesion is able to grow, cells may gain additional genetic and/or epigenetic aberrations and eventually a malignant tumor may be formed. Even if cellular senescence is induced, cells can potentially bypass the senescent state (e.g., by epigenetic changes, etc.) and undergo malignant transformation [25,26,31,33].

Although cellular senescence is involved in normal development and ensures tissue homeostasis by limiting the growth of damaged cells, it can have detrimental effects as well. With aging, there is an accumulation of senescent cells resulting in tissue aging and eventually failure of organ homeostasis and function. Cellular senescence has been recognized as an important hallmark of aging [25,31,32]. This has been demonstrated by Tyner et al. [34], who performed a mice experiment evaluating P53 function. As explained above, P53 is a key tumor suppressor and an inducer of cellular senescence. The experiment compared P53 wild type, P53 knockout and P53 mutant mice. Mutant mice had a mutation in *TP53* gene by which P53 was activated rather than inactivated. More than 45% of wild type mice and over 80% of knockout mice developed large tumors of various types. By contrast, mutant mice exhibited an increased tumor resistance since only 2 of 35 mice developed a localized tumor lesion [34]. However, compared to the other strains, the mutant mice showed an early onset and enhanced aging phenotype, resulting in reduced lifespan [34]. Noteworthy, telomere shortening has been acknowledged as an activator of P53 [35]. García-Cao et al. [36] studied the impact of P53 on the elimination of telomere-damaged cells and/or on telomere-driven aging. They saw that the number of telomere-damaged cells was lower in mice carrying extra copies of *P53* compared to wild type mice, confirming the role of P53 in the elimination of telomere-damaged cells. Furthermore, an aging-promoting effect of increased P53 activity on telomere-driven aging could not be established [36]. Similar observations have been made for P16^INK4a^: knocking out *P16^INK4a^* in mice resulted in an increased frequency of cancer development and was associated with shorter survival compared to wild-type mice [37,38]. Furthermore, elephants carry extra copies of *TP53* gene, by which they are more likely to generate an apoptotic response in damaged cells, which clarifies their tumor resistance [39]. These observations confirmed the important cancer protective role of P53.

Additionally, apart from cell cycle arrest, senescent cells exhibit a SASP, characterized by the secretion of numerous inflammatory mediators i.e., cytokines and chemokines such as IL-1α, IL-6, IL-8, and Interferon gamma (IFNγ), proteases like matrix metalloproteinases (MMPs), Cathepsin B and growth factors such as vascular endothelial growth factor (VEGF) and insulin-like growth factor (IGF)-binding proteins [31,32]. SASP often has several positive functions in the short term but these can become detrimental in the long term, by promoting both the aging process and tumor development, as explained below [31,32,40]. Various effects of the SASP are shown in Figure 4 [32].

When senescent cells are temporarily present in damaged tissue, the SASP ensures improved wound healing. However, in the long term, senescent cells and their SASP can affect tissue structure and function, which may contribute to the aging process e.g., by the secretion of MMPs. MMPs are also associated with tumor cell invasion and migration in many cancers. Furthermore, the SASP component VEGF can stimulate tumor growth by promoting angiogenesis and tumor cell invasion. Additionally, cell proliferation is enhanced by IL-6, IL-8 and MMPs. Senescent cells reinforce their own senescent phenotype via SASP, but SASP factors can also induce senescence in neighboring cells (e.g., IL-1α, transforming growth factor beta (TGF-β) and IL-6) [31,32,40]. Another crucial function of cytokines and chemokines included in SASP is the recruitment of immune cells; these ensure clearance of senescent cells and enhance the local immune response against a developing tumor (e.g., macrophages, natural killer (NK)-cells, T-cell, etc.) [31,32,40]. However, immunosuppressive cells are attracted as well (e.g., myeloid derived suppressor cells). Importantly, SASP also comprises the secretion of numerous inflammatory cytokines (e.g., IL-1α, IL-6, IFNγ, TGF-β), chemokines (e.g., IL-8, monocyte chemoattractant protein (MCP)) and other mediators (e.g., granulocyte/macrophage colony-stimulating factor, macrophage inflammatory proteins) [31,32,40]. It is likely that “inflammaging”, the low-grade, chronic state of inflammation observed in the aged, is partially generated by the SASP of senescent cells accumulating in the body. Chronic inflammation has several detrimental effects: it modifies the microenvironment and modifies functions of surrounding cells; it affects anabolic signaling (e.g., decrease of IGF-1 by IL-6 and tumor necrosis factor alpha (TNF-α)); it influences immune reactions. Mortality and many age-related diseases, such as cardiovascular diseases, diabetes, cancer, and even frailty, are associated with inflammation [4,31,32,40].

### 2.3. Immunosenescence

A functional immune system is highly important in the prevention of tumor growth. However, aging leads to a progressive decay of immune functions, referred to as immunosenescence. Due to immunosenescence, an effective immune response against a developing tumor may fail. Moreover, it has been considered as one of the leading causes of many other age-related diseases. A strong interconnection between immunosenescence and inflammaging exist, both appear to maintain and induce each other [41,42].

#### 2.3.1. Innate Immunosenescence

Although aging mainly alters the adaptive immune system, the innate immune system is reshaped as well (Figure 5) [7,16,41,42]. Noteworthy, the innate immune system is activated by pathogens but also by an inflammatory response. In addition, cells of the innate immune system are able to release inflammatory cytokines and chemokines, highlighting the link with inflammaging [41]. With aging, the number of neutrophils remains relatively stable. However, neutrophil function is altered, as demonstrated by reduced chemotaxis, phagocytosis, signaling, and increased susceptibility to apoptosis [7,16,42]. There are age-related shifts within the monocyte subpopulation, for instance the proinflammatory monocytes (CD16^+^) increase and the phagocytic monocytes (CD16^−^) decrease. This is accompanied by many functional changes of monocytes/macrophages e.g., reduced chemotaxis, phagocytosis, antigen presentation, and increased production of inflammatory cytokines [7,16,42]. In dendritic cells, antigen presentation and signaling are reduced, and myeloid dendritic cells produce more proinflammatory cytokines [7,42]. The total number of NK-cells increases with aging, and shifts in the subpopulations are seen i.e., a decrease of the immature, immunoregulatory (CD56^bright^) and an increase of the mature, cytotoxic NK-cells (CD56^dim^). As a result, response to cytokines and signaling is impaired, and the NK cells produce less cytokines and become less cytolytic [7,16,42]. The altered functions of innate immune cells with aging may have an effect on the adaptive immune system, which is activated when an antigen, mostly presented by antigen presenting cells of the innate immune system, is recognized by T- and/or B-cells [16,41].

#### 2.3.2. Adaptive Immunosenescence

The adaptive immune system is composed of T-cells, mainly mediating cellular immunity, and B-cells, which play a major role in antibody-mediated or humoral immunity [42]. With aging, shifts in both numbers and functions of adaptive immune cells occur (Figure 5) [7,17,41,42]. In general, immunosenescence of the adaptive immune system is characterized by a decrease of naive cells and an accumulation of memory cells. These changes are most striking in the T-cell population [7,41,42], with more pronounced age-related alterations in the cytotoxic T-cells (CD8^+^) compared to the helper T-cells (CD4^+^) [17]. With aging, pluripotent hematopoietic stem cells (HSC), from which all immune cell types originate, are more likely to differentiate into myeloid than lymphoid cell lines [7,41,42]. More importantly, however, is the thymic involution after puberty, which results in a reduced production of naive T-cells and lowered T-cell receptor (TCR) diversity [7,17,41,42]. Moreover, the proliferative capacity diminishes and a lifelong exposure to numerous immune challenges further reduces TCR diversity and depletes the naive T-cell reserves by conversion into memory T-cells [7,17,41,42]. Various studies have demonstrated the alterations in the T-cell compartment by evaluating different subset markers. Aging is associated with an alteration of the CD4/CD8 ratio and expansion of the immunosuppressive regulatory T-cells (Tregs) [41,42]. Naive, central memory (CM), effector memory (EM) and terminally differentiated effector memory (TEMRA) cells are identified by assessing the expression of CD45RA and CC-chemokine receptor 7 (CCR7). The costimulatory receptors CD27 and CD28 are important markers of T-cell activation, and CD57 and killer lectin-like receptor G1 (KLRG1) are markers of senescence [17,42]. With age, Koch et al. saw a decrease in naive T-cells (CD45RA^+^, CCR7^+^) and an expansion of TEMRA T-cells (CD45RA^+^, CCR7^−^), which was more pronounced in the CD8^+^ as compared to the CD4^+^ compartment [17]. Additionally, the expression of the costimulatory receptors CD27 and CD28 declined, especially in CD8^+^ T cells, and cells with lower expression of CD27 and CD28 often had a higher expression of CD57 and KLRG1. Hence, with age, numbers of naive T-cells decrease while the frequency of late-stage differentiated/senescent memory T-cells increases [17]. Apart from an increase in the expression of senescence markers, aging can also be associated with an increase in the expression of markers of exhaustion, which include inhibitory immune checkpoints (e.g., programmed cell death protein 1 (PD-1), cytotoxic T-lymphocyte-associated protein 4 (CTLA-4), etc.) [7].

Although age-related changes in the B-cell population are less apparent, certain modifications could be identified as well. Like T-cells, B-cells originate from HSC, thus the age-related shift from lymphopoiesis to myelopoiesis also affects the B-cells. There is also a decrease in pre-B-cells and an increase in memory B-cells, the diversity and production of antibodies diminishes which eventually leads to altered B-cell functions [42].

#### 2.3.3. Clinical Implications of Immunosenescence

As expected, immunosenescence can have important clinical implications. There is an increased susceptibility to infections, partly due to impaired innate immune cell functions and the decrease of naive T-cells [41,42,43]. In older adults, vaccination has been shown to be less efficient, which may be a result of the decrease in naive T-cells and to the lowered antigen presenting capability of dendritic cells [41,42,43]. Importantly, as mentioned before, inflammation associated with immunosenescence creates a protumor microenvironment, which together with the lower capacity to recognize neoantigens, altered immune function, and lower T-cell activation, may lead to a decreased antitumor response [7,41,42,43].

Noteworthy, immunotherapy mainly depends on a properly working immune system. Immunosenescence may have an important impact on the efficacy of cancer immunotherapy but also on its side effects [7]. As mentioned before, naive T-cell numbers decrease with age, memory T-cells numbers increase, and these memory cells are often senescent or exhausted [17]. Immune checkpoint inhibitors, used in immunotherapy, block the interaction between inhibitory receptors and their ligands resulting in reactivation of the immune cell. With age, expression of these inhibitory receptors (e.g., PD-1, CTLA-4), as well as senescence markers (e.g., CD57) increase, while costimulatory receptors (e.g., CD28) are lost. This might have an impact on reactivation of the T-cells, making it less effective in older adults [7]. However, in melanoma, Kugel et al. [44] observed a higher CD8/Treg ratio in the tumor, which corresponded with a higher response to anti-PD-1 therapy (Pembrolizumab) in older patients (>60 years) [44]. Importantly, it should be noted that in general, in contrast to what was seen in this study, aging is associated with an increase of Tregs [41]. Side effects associated with immune checkpoint inhibitors are often related to inflammation. As inflammation already increases with aging, the inflammatory toxicities might be more severe in older patients [8]. The underrepresentation of older patients with cancer in clinical studies is also an issue in the immunotherapy field. In addition, the few older patients who are included in trials are usually selected fit persons. The more frequent frail old population is very rarely well represented in clinical trials with new drugs. Little is known about the immune checkpoint immunotherapy efficacy or tolerability in the general older population and certainly in the more frail subgroup [10]. As of yet, literature shows inconsistent results when comparing younger and older patients treated with immune checkpoint inhibitors. Some studies show similar results in terms of efficacy and tolerability for old patients compared to younger patients while others show a lower efficacy in old patients and an increased risk of side effects [6,7,8]. However, none of these studies particularly focused on older (and certainly not frail) adults, thus the numbers of older patients included in these trials are often very small and the results should be interpreted with caution [6,7,8].

Apart from immune checkpoints, cancer antigens can be targeted by immunotherapy as well. There are two major classes of cancer antigens: tumor specific-antigens (TSA) and the tumor-associated antigens (TAA). TSAs originate from oncogenic mutations and are therefore exclusively found on tumor cells. These antigens have gained interest over the past years as they are highly cancer-specific and are therefore associated with a lower risk of side effects [7,45]. The immune system has never been exposed to these neoantigens and with the shrinkage of the naive T-cell pool associated with immunosenescence, immunotherapies targeting these antigens might be less effective in older patients. Nonetheless, immunotherapies targeting TAAs might be more effective in older patients. TAAs are self-antigens that are overexpressed on tumor cells and are a result of tissue differentiation. They are currently widely used as cancer immunotherapy targets, an example in breast cancer is Human epidermal growth factor receptor 2 (HER2) [7,45]. The immune response against TAAs mainly depends on memory cells that have already been exposed to these antigens. Nevertheless, although there is an increase of memory cells with age, the proliferation capacity and activation might be lower. Moreover, TAAs are less tumor specific and are associated with higher risk of side effects compared to TSAs as they are also found on normal cells throughout the whole body [45]. Additionally, other options in older patients might be anti-Treg therapy, stimulation of the innate immune system, chimeric antigen receptor T-cell therapy, NK-cell adoptive therapy, etc., however no data is available as of yet and the immunologic changes in older patients with cancer should be investigated more thoroughly [7].

Importantly, the age-related changes in the immune response could have substantial effects on the amount, composition and distribution of the tumor immune infiltrate [46]. First, aging might have an impact on the ability of immune cells to recognize and respond to a developing tumor. Secondly, the composition of the immune infiltrate might be altered, as there are numerous age-related shifts within the immune cell populations. Thirdly, the capacity of immune cell to infiltrate into the tumor may be affected, resulting in altered spatial distribution of the immune cells within the tumor. However, thus far, the effect of aging on the tumor immune infiltrate has only been poorly studied. By performing a detailed characterization of the tumor immune infiltrate in patients with cancers, important insights may be gained. Markers worthy of investigating are, for example, the stromal tumor infiltrating lymphocytes (sTILs), as they are easy to measure and are informative of immune cell abundance in the tumor. Moreover, composition of the immune infiltrate should be investigated as well by assessing the expression of different immune cell markers. In this regard the use of multiplexed immunohistochemistry will allow us to get a better understanding of the tumor associated immune infiltrate elucidating the significance of the coexpression of different markers in a specific cell and its relation with other immune cells or with tumor cells [47,48]. For instance, CD3 (T-cells), CD4 (helper T-cell), CD8 (cytotoxic T-cells), CD20 (B-cells), FOXP3 (Tregs), etc. Additionally, activation (CD27, CD28), ICs (PD-1, CTLA-4, lymphocyte-activation gene 3 (LAG-3), T-cell immunoglobulin, and mucin domain-containing molecule 3 (TIM-3)) senescence (CD57, KLRG-1), exhaustion (PD-1 and/or CTLA-4) markers could be included as well [7]. Furthermore, the spatial distribution of the immune cells has been shown to be a prognostic factor [49].

Thus, immunosenescence and inflammaging are significant consequences of the aging process and are probably the most important causes of many age-related diseases, including cancer. However, as of yet, the clinical relevance of immunosenescence and inflammaging is underexplored, especially in the cancer setting. In order to gain insights, our group performed an extensive immune biomarker study in patients of different ages (35–45, 55–65, ≥70 years old) with so called luminal B-like breast cancers [46]. Numerous profound age-related alterations, which could be linked to inflammation and/or immunosenescence, were observed in the blood immunological portrait and the local immune response. The plasma levels of several inflammatory mediators (IL-1α, IP-10, IL-8, MCP-1, and C-reactive protein (CRP)), immune checkpoint markers (Galectin-9, sCD25, TIM-3, and Programmed death-ligand 1 (PD-L1)), IGF-1 and circulating immune-related microRNAs (miRNAs) (miR-18a, miR-19b, miR-20, miR-155, miR-195, and miR-326) significantly changed with aging. Moreover, within peripheral blood mononuclear cell populations, several shifts were seen, and the population of naive cytotoxic T-cells showed to be most affected by aging. With increasing age, the tumor immune infiltrate changed as well. The total lymphocytic infiltration was lower in older patients, concomitant with lower abundance several immune cell populations (T-cells, cytotoxic T-cells and B-cells). The relative fractions of cell subsets in the immune infiltrate were also altered. In the old group, frailty-related changes were studied as well. Here, the most striking observations were the increase in exhausted/senescent (CD27^−^CD28^−^ and/or CD57^+^) terminally differentiated cytotoxic T-cells in blood and an increased infiltration of FOXP3^+^ cells in the tumor. This data confirmed an age/frailty-related remodeling of both systemic immunity and the tumor immune response in patients with a luminal B-like breast cancer [46].

### 2.4. Treatment of Older Patients with Cancer

In several solid cancer types, tumor behavior differs with age. For example, older patients are more frequently diagnosed with a more aggressive ovarian cancer compared to younger patients while the opposite is true for breast cancer patients [11]. Importantly, at old age tumors are often more advanced at the time of diagnosis [50]. The highly heterogeneous global health status of older people and the fact that older cancer patients are highly underrepresented in clinical trials, makes elderly cancer care highly complex [10,50]. Even if older patients are included in clinical studies, results should be interpreted with caution; older patients who are included in clinical studies are often very fit and are therefore not representative for the general older population [10].

Thus, it is important to take into account a patient’s biological age rather than merely chronological age. A comprehensive geriatric assessment (CGA) could be used to assess a person’s biological age by determining the fitness/frailty status. This multidimensional and interdisciplinary tool evaluates several health domains via standardized questionnaires, such as functionality, mental health and social status, comorbidities, cognition, polypharmacy, and nutritional status. These elements could have an important impact on cancer development, prognosis, treatment effectiveness, and/or tolerability [11]. Declined functional status could be linked with increased mortality [11,51]. Comorbidities can, by themselves and/or by their specific treatments, increase the risk of developing a cancer. For instance, tuberculosis is associated with lung cancer development, and chronic inflammation and immunodeficiency are linked with several cancers [52] and statins correlated with a lower risk of developing breast, large bowel and prostate cancer [53]. Furthermore, comorbidities might affect treatment tolerability [52] and drugs administered to treat the comorbidities might interact with anticancer therapies [11]. Nutrition can also be linked with cancer, life expectancy and treatment tolerability [11,26]. A very important factor to take into account is dementia, which has an important impact on treatment decision: not only can the correct intake of medication be impaired but several anticancer therapies could also have neurotoxic effects (e.g., aromatase inhibitors blocking estrogen production) [54,55]. Thus, the frailty status of older patients but not age by itself, should be a determining factor to give a certain treatment or not. Several studies also demonstrated that frailer patients undergoing surgery are confronted with worse survival and might have more postoperative complications [56]. Independent of age, frailty status was associated with a worse overall survival and disease-specific survival in patients with breast cancer after tumor resection [57]. Overall survival of frailer patients with colorectal cancer who had a colectomy was also less favorable [58]. Tolerability of and survival after chemotherapy might also be affected by a patient’s frailty status [10,56]. Amongst many others, Hurria et al. established a correlation between frailty and the risk of toxicity during/after chemotherapy [59]. In breast cancer, frailty was associated with both increased toxicity and worse survival [60]. Conversely, limited literature is available on the effects of radiotherapy on frailer patients, although it is obvious that frailty might play a role here as well [56]. The potential impact of immunosenescence on the effectiveness of cancer immunotherapy has been described extensively in the previous section.

Ensuring healthy aging could be an important approach not only to prevent/treat cancer but also other age-related diseases [61,62,63]. Several factors could contribute to healthy aging. For example, caloric restriction resulted in an increased lifespan and a delay of age-related diseases in different animal models [26,61]. In humans, it also showed health benefits (e.g., reduced cardiac risk factors) [26]. The importance of the insulin signaling pathway [61,64,65,66,67,68,69,70], mechanistic target of rapamycin (mTOR) [61,69], sirtuins [61,71], circadian clock [61,72], and oxidative stress [61,73] in relation with longevity and/or age-related diseases have been extensively documented. As cancer and other age-related diseases are linked with inflammation, maintaining a good balance between pro- and anti-inflammatory mediators might be of great importance at higher age [61]. Furthermore, the use senolytic drugs (i.e., drugs eliminating senescent cells) for the treatment of senescence-associated cancers are gaining interest [61,62,63]. Data from clinical studies in humans are not available yet, however, studies in mice and human cells were very promising. For instance, Amor et al. studied chimeric antigen receptor (CAR) T-cells as a senolytic agent by targeting plasminogen activator receptor on senescent cells in mice. This resulted in better survival in mice with a lung adenocarcinoma and in mice with liver fibrosis, tissue homeostasis was restored [62]. In another study in mice, senescent cells were cleared via recombinant caspase 8 which was activated in senescent cells after treatment with the AP20187 drug, resulting in delayed onset of age-related disorders [74]. However, further research is mandatory and it should be kept in mind that senescent cells also have several beneficial effects, which may outweigh the benefits of being removed [63].

## 3. Aging Biomarkers

As the role of aging is largely underexplored in the cancer field so far, it is important to gain more extensive and in-depth insight in the aging process in the oncological setting. This could be achieved by studying various aging biomarkers that have been described in literature in the healthy population, some of which are listed below [64,75]. Additionally, as the immune system is highly affected by aging and as it is an important factor in cancer development but also in cancer immunotherapy, markers of immunosenescence are also highly relevant in a cancer context. Moreover, a better understanding of the impact of aging on tumor immunity is mandatory, especially in view of increasing clinical successes of cancer immunotherapy. Here below, we summarize some of the most studied and relevant aging biomarkers in the general older population that could be relevant for future research in cancer patients.

### 3.1. Gene Expression

Age-related changes of gene expression contribute to the physiological alterations observed with human aging and could thus be studied as well [12,64]. Vo et al. compared the abundance of a selection of 148 transcripts involved in immunosenescence and stress response in PBMC from healthy young, middle aged and old persons and found 16 differentially abundant transcripts that can be considered as easily accessible biomarkers of aging [21]. Some of these age-related genes are involved in T-cell function (e.g., CD28, CD69, lymphocyte protein tyrosine kinase (*LCK*): decreased abundance in old subjects), in inflammatory reactions and in oxidative stress response (e.g., tumor necrosis factor receptor superfamily member 1A (*TNFRSF1A*), heme oxygenase 1 (*HMOX1*) and heat shock protein family A member 6 (*HSPA6*): increased abundance in old subjects). This is in agreement with the T-cell hypo-responsiveness and the low-grade proinflammatory status observed in old persons and could thus contribute to the lack of appropriate (tumor) immune responses in older compared to younger persons [21]. Peters et al. identified 1497 genes in whole blood with a differential expression with chronological age. Moreover, they were able to identify several cluster of genes, which could be linked to different age-related alterations e.g., immunosenescence, altered RNA metabolism functions, etc. [76]. Several other genes have been linked to the aging process, such as low density lipoprotein receptor-related protein 1B (*LRP1B*), paraoxonase 1 (*PON1*), ‘ataxia telangiectasia mutated’ (*ATM*) gene, *p21/CDKN1A* gene, p53 protein, insulin/IGF-1 signaling (IIS) components, telomerase RNA component (*TERC*), *IL-1* gene, *IL-6 gene*, Toll-like receptors genes (*TLR*) [77].

### 3.2. Single Nucleotide Polymorphism

Genetic predisposition plays a determining role in healthy longevity vs. frailty development. A single nucleotide polymorphism (SNP) is a DNA sequence variation at a single nucleotide position. With regard to aging, a specific SNP within the forkhead box O3 (FOXO3a) gene has been associated with longevity and SNPs in apolipoprotein E (APOE) have been correlated with an age-associated phenotype [78,79].

### 3.3. DNA Methylation Profiles

Altered DNA methylation also appears to be associated with the aging process [64,80]. These altered DNA methylation patterns are caused by numerous extrinsic factors e.g., lifestyle, environmental factors. DNA methylation mostly alters cytosine nucleotides followed by a guanine nucleotide, called CpG dinucleotides [81]. CpG methylation of the genome ensures genomic stability and it acts as a repressor of DNA recombination at telomeres [81,82]. Additionally, methylation of CpG sites in the promoter of a gene may inhibit gene expression, making DNA methylation an important player in gene regulation and “silencing”. Interestingly, age-related hypomethylation of the global genome is well-documented, while, on the other hand, many specific CpG sites are subject to age-related DNA hypermethylation. Several age-dependent CpG signatures have recently been reported as “DNA methylation age” biomarkers that can predict all-cause mortality [83]. Weidner et al. showed that assessing DNA methylation at three age-related CpG sites could predict age [80].

### 3.4. Telomere Attrition

Telomeres are complex, repetitive regions consisting out of (TTAGGG) repeats and associated proteins. With each cell division, and thus with increasing age, telomeres that ensure genomic integrity by protecting the end of DNA chromatids are shortened, and when becoming critically short they can induce cellular senescence [25]. Leukocyte telomere length has been suggested as a biomarker of biological aging [12,64]. In a study of Cawthon et al. shorter telomere length was associated with worse survival due to higher rates of heart and infectious diseases [22]. Furthermore, Mons et al. demonstrated that telomere length was associated with age and all-cause mortality [84].

### 3.5. Oxidative Stress Markers

One of the mechanistic explanations of the aging process is the oxidative stress/free radical theory. Due to metabolic processes and mitochondrial respiration, reactive oxygen species are produced, which exist in the cell in equilibrium with antioxidant molecules [28,29]. Oxidative stress increases with age, leading to accumulation of oxidation products of lipids (e.g., isoprostanes), nucleic acids (e.g., RNA derived 8-hydroxyguanosine or DNA derived 8-hydroxy-2′-deoxyguanosine) and proteins (e.g., nitrotyrosine), that exert deleterious effects and ultimately cause cellular dysfunction [64]. Isoprostanes, 8-hydroxyguanosine and 8-hydroxy-2′-deoxyguanosine can be measured as aging biomarkers in plasma [85]. Simonek et al. reported an increase of 8-hydroxy-2′-deoxyguanosine with aging [86], and in a study of Montine et al. the level of isoprostanes increased with aging [87].

### 3.6. Plasma miRNA Profiles

miRNAs are small noncoding RNAs which are 21–25 nucleotides in length. They mediate post-transcriptional gene silencing through either translational repression or targeting of the messenger RNA for degradation. They play a critical role in numerous physiological pathways, developmental processes and pathological conditions [64]. Dysregulation of specific miRNAs is associated with several disorders (e.g., cancer, cardiovascular diseases, etc.). It is believed that miRNAs also play a role in aging, immunosenescence and inflammation but also frailty [64,88,89,90,91,92]. By consequence, miRNA signatures could also be interesting as potential aging/immunosenescence/inflammation markers, especially since plasma miRNAs have recently been recognized as sensitive, specific and extremely stable biomarkers [64,88,89]. For instance, miR-20a and miR-181a have been reported as aging associated plasma miRNAs [64,88]. The miR-17–92 cluster and miR-155 are highly involved in immunological processes [89,90] and important inflammatory miRNAs are miR-155 and miR-21 [64,91]. Additionally, the expression levels of particular miRNAs seem to correlate with clinical frailty, e.g., miR-92a and miR-326 [92]. In our own biomarker study six immune-related plasma miRNAs showed different expression levels between different age groups. These included three members of the miR-17–92 cluster namely miR-18a, miR-19b and miR-20a. miR-18a decreased with age whereas miR-19b and miR-20a peaked in the middle-aged group but had the lowest expression in the oldest age group. The latter was also true for miR-195, which is associated with T-cell activation. Noteworthy, the miR-326 could only be measured in the oldest age group [46].

### 3.7. Proteostasis

Aging as well as several age-related diseases (including cancer) are associated with loss of proteostasis due to impairment of chaperoning, proteasome activity and autophagy–lysome activity [25,26]. However, these systems are rather activated in a cancer context [26], making interpretation of changes in these markers very difficult in older patients with cancer.

### 3.8. Markers of Inflammation

Aging has been associated with a low-grade, chronic state of inflammation, also referred to as inflammaging. Apart from the above-mentioned miRNAs, inflammation can be assessed by measuring plasma levels of numerous inflammatory mediators. Several reports have described a gradual increase in circulating proinflammatory cytokines (e.g., IL-1, IL-6, TNF-α), chemokines (e.g., IL-8, MCP-1) and other inflammation markers (e.g., CRP), concomitant with a decrease in anti-inflammatory mediators (e.g., IL-10) with increasing age. Circulating levels of inflammatory mediators are often elevated prior to cognitive decline, dementia and loss of physical performance [75,93,94]. At the cellular level, these markers of inflammation are also associated with cellular senescence, more particular with the SASP. In our cancer cohort, a clear increase of several inflammatory mediators was observed. With aging higher plasma levels of inflammatory cytokines (IL-1α), chemokines (IP-10, IL-8, MCP-1) and the clinical inflammation biomarker CRP were seen [46].

### 3.9. T-Cell P16^INK4a^ Expression

In our previous biomarker study, an increase of T-cell *P16^INK4a^* expression with increasing age was noted in patients with luminal B-like breast cancer [46]. Cellular senescence is one of the most important hallmarks of aging [25]. The expression of the senescence marker *P16^INK4a^* in T-cells has been described as a biomarker of lymphocyte senescence and thus aging of the immune system. In healthy humans, *P16^INK4a^* expression increases markedly in peripheral blood T-cells with physical inactivity and exposure to mutagens such as tobacco. In addition, the expression of *P16^INK4a^* increases with chronological age, with an average 10-fold increase between the ages of 20 and 80 [19].

### 3.10. Shifts in Immune Cell Subpopulations

As already pointed out above, numerous age-related changes occur in the cellular components of the innate and even more pronounced in the adaptive immune system [16,17]. These changes can be demonstrated by flow cytometric subtyping of the different immune cell populations [17]. Interesting markers to include in such analysis are CD14 and CD16 to distinguish different monocyte subpopulations, and CD56 and CD16 to identify different NK subpopulations [16]. CD4 and CD8 can identify the helper T-cells and cytotoxic T-cells of the adaptive immune system. A distinction between naive and memory cells can be made with the help of CD45RA and CCR7 expression. Moreover, activation status can be assessed by evaluating CD27 and CD28 whereas CD57 is a marker of senescence [17]. Koch et al. analyzed CD4 and CD8 T-cell subsets via flow cytometry and observed a decrease of naive cells and an increase of late-differentiated T-cells, especially those expressing CD8 [17]. Sadeghi et al. showed higher levels of inflammatory monocytes (CD14^dim^CD16^bright^) with age [95] and Chidrawar et al. demonstrated that CD56bright NK-cell abundance decreased with increasing age [96]. In our breast cancer cohort, we observed an increase of intermediate monocytes with age. However, most notable were the effects of age on the naive CD8 population. Within the CD8 population we also saw a loss of CD27 and/or CD28 [46].

### 3.11. Markers of Cellular Senescence

At the cellular level, the SASP as well as elevated P16, P53 and P21 levels are described as important markers of cellular senescence. Additionally, increased levels of the DNA damage marker γ-H2AX, higher degree of β-galactosidase activity and the formation of high levels of senescence-associated heterochromatin foci characterize senescent cells [25,64,97].

### 3.12. Circadian Clock

With age, the circadian rhythms change, which has an important impact on several biological processes. For example, older adults’ sleeping patterns are different, which may increase the risk of cognitive decline [72,98]. Melatonin levels decrease with aging and may be associated with age-associated cognitive disorders [72]. Numerous metabolic process are regulated by circadian clocks such as glucose homeostasis, lipid metabolism, as well as inflammation [72]. Moreover, a connection between deregulation of the circadian clock and cancer has been described [99,100]. Several studies showed that sleep disruption increased the risk of developing e.g., breast [101] and prostate cancer [102]. Conversely, optimizing circadian rhythms might improve outcome, e.g., cortisol levels were associated with lung cancer survival [103].

### 3.13. IGF-1

One of the hallmarks of aging is dysregulated nutrient sensing with an important role for the insulin/IGF-1 signaling pathway [64]. IGF-1 can activate the insulin receptor but also has important growth stimulating effects. The plasma levels of circulating IGF-1 have been shown to decrease with aging. This was also confirmed in our cancer cohort [46]. Diminished IGF-1 signaling has been associated with longevity in several mouse studies [65,66], some studies in humans could confirm this, however, results are often inconsistent [65,66,67,68]. Noteworthy, reduced IGF-1 signaling has been linked to an improved functional status [67]. Studies in mice but also some studies in older humans showed that diminished IGF-1 signaling has been associated with an improved functional status. Moreover, although the link between IGF-1 levels and age-related diseases is not fully established, it has been demonstrated that increased levels of IGF-1 are associated with tumor development, whereas it might reduce the risk of cognitive decline, dementia and cardiovascular disease [67,69,70].

### 3.14. Microbiome

The gut microbiome might be an important factor in healthy aging as it has been acknowledged as an important regulator of metabolic processes, immunity and inflammation [25,104]. With aging the microbiota composition becomes less diverse and there are changes in the abundance of specific species [104]. Analyzing the gut microbiome might give important information; for example, Biagi et al. showed that extreme longevity was associated with an increased abundance of *Christensenellaceae, Akkermansia* and *Bifidobacterium* [105]. Moreover, frailty seems to be associated with a reduced microbiota diversity [106]. Studies in mice already showed that modulation of the microbiome might delay immunosenescence. In humans Shen et al. demonstrated that decreased levels of *Bacteriodetes* might be linked to immunosenescence [107]. Additionally, the microbiome association between cancer development and anticancer therapies have been made [108]. *Helicobacter pylori* is a widely acknowledged oncogenic organism [109] and several studies show a relation between treatment response and the gut microbiome [108].

### 3.15. Biomarker Panels

Recently, we evaluated blood immunosenescence signatures potentially reflecting age or frailty. This was done by assessing individual biomarkers as well as 3-biomarker panels and their ability to divide patients into the correct age/frailty group. This study revealed that blood biomarker panels performed better than individual biomarkers, and that these panels did not only accurately reflect a patient’s chronological age, but more importantly the patient’s frailty status (manuscript submitted). These findings confirmed the potential utility of biomarkers in geriatric oncology.

## 4. Conclusions

The link between aging and cancer has been widely acknowledged, a schematic overview can be seen in Figure 6. The lifelong exposure to numerous endogenous as well as exogenous factors, the chronic state of inflammation and immunosenescence contribute to an increased risk of developing cancer with advancing age. However, the biological age of each individual is unique and is not reflected by their chronological age but rather by frailty status. Here we described several potential aging biomarkers that may provide more in depth view of the aging/frailty status as compared to clinical interpretation only, the implementation of aging biomarkers or a combination of them in clinical practice could aid with the determination of a person’s true age and therefore with (oncological) decision-making. Moreover, as aging has important effects on several biological process, the effectiveness and tolerability of certain treatments (e.g., cancer immunotherapy) could be altered in older patients. Thus, selecting the most suitable treatment for older patients is highly important yet extremely difficult. This warrants further and more extensive investigation in these older patients. Better clinical insight of older patients and the identification of robust, reliable biomarkers could aid with a better individualized treatment for older patients with cancer.

## Figures and Tables

**Figure 1 cancers-13-01400-f001:**
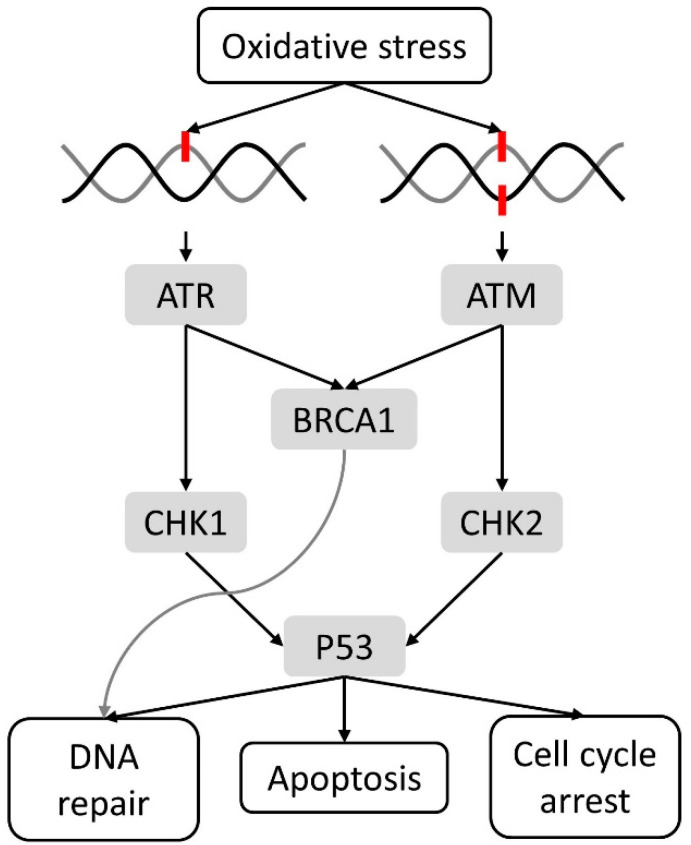
Simple overview of the DNA damage response after oxidative DNA damage. Double strand breaks recruit ataxia-telangictasia-mutated (ATM), whereas single strand breaks induce ataxia-telangictasia-Rad3-related (ATR). DNA damage response mediators and downstream kinase can be activated whereby either DNA repair, apoptosis or cell cycle arrest will occur.

**Figure 2 cancers-13-01400-f002:**
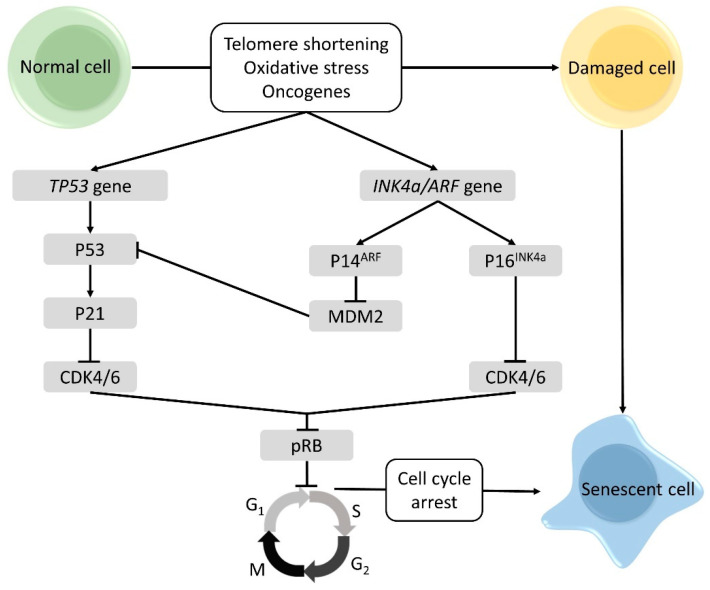
Pathways of cellular senescence. Due to cellular stressors like telomere shortening, DNA damage, oxidative stress and the expression of oncogenes, senescence genes are induced. The *TP53* gene encodes for P53, which induces P21, a cyclin dependent kinase inhibitor (CDK) blocking CDK4/6. As a result, phosphorylation of the retinoblastoma protein (pRB) is hindered. Consequently, the cell cannot enter the S-phase and the cell cycle is arrested in the G1 phase. The *INK4a/ARF* locus encodes for P14^ARF^ and P16^INK4a^ protein. P14^ARF^ is a regulator of P53 activity: by binding mouse double minute 2 homolog (MDM2), degradation of p53 is avoided. Like P53, P16^INK4a^ represses CDK4/6 by which pRB is not phosphorylated and cell cycle progression is prohibited. Figure adapted from [32].

**Figure 3 cancers-13-01400-f003:**
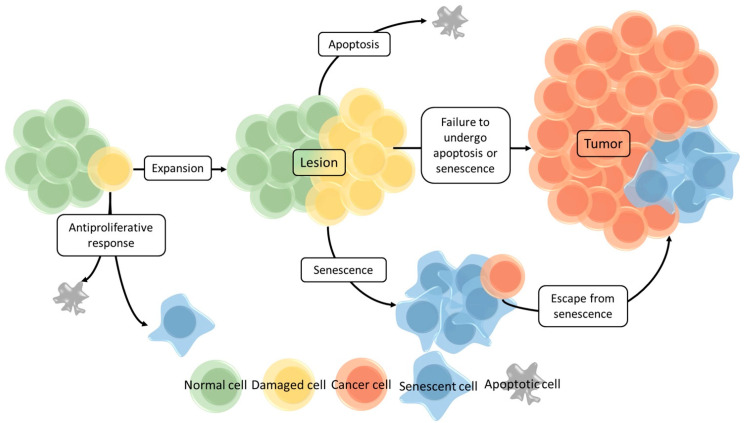
Schematic overview of antitumor protection by cellular senescence. Damaged cells can become apoptotic, enter the state of senescence (antiproliferative responses) or continue to replicate (expansion). When the latter occurs, a lesion may form where cells, again, can become apoptotic or enter the state of senescence. If suitable defense mechanisms are absent or fail, the lesion can further expand and by gaining additional mutations, a cancerous lesion (tumor) may be formed. Moreover, senescent cells still can escape this state and become cancerous as well. Normal cells are indicated in green, damaged cells in yellow, cancer cells in red, senescent cells in blue, and apoptotic cells in grey. Reprinted with permission from ref. [33]. Copyright 2021 American Pharmaceutical Association.

**Figure 4 cancers-13-01400-f004:**
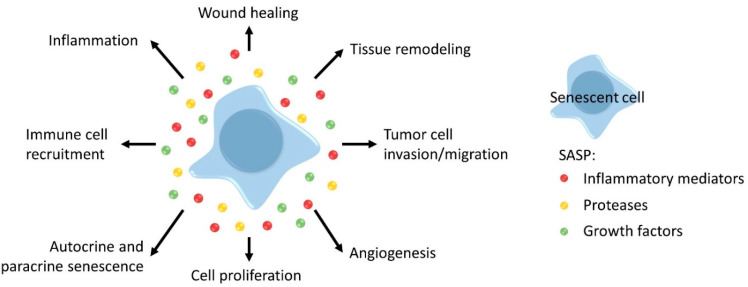
Overview of the important SASP functions. SASP components induce wound healing, tissue remodeling and recruit immune cells. Protumor effects of the SASP are tumor cell invasion and/or migration, stimulation of vessel formation, induction of cell proliferation, and the development of an inflammatory environment. Different SASP factors establish autocrine and paracrine senescence. Figure adapted from [32].

**Figure 5 cancers-13-01400-f005:**
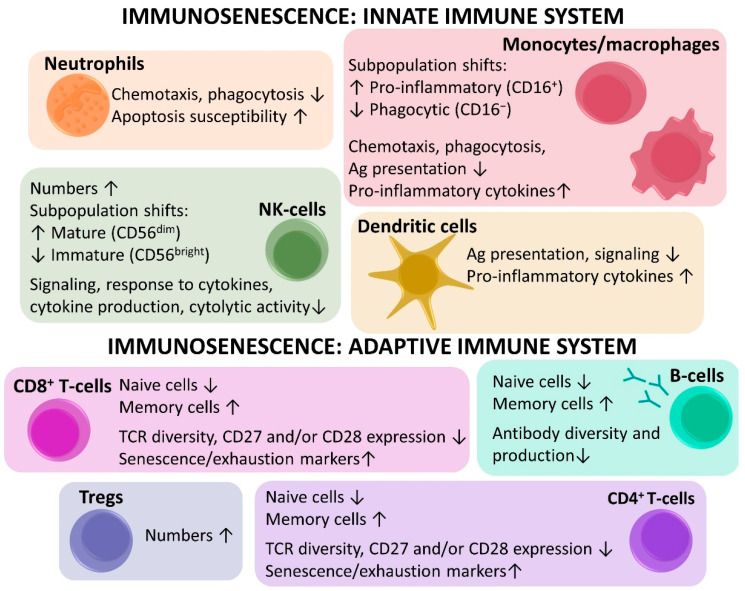
Overview of the age-related changes in the innate immune system, including neutrophils, monocytes, NK-cells, dendritic cells, and adaptive immune system with the cytotoxic CD8^+^ T-cells (most affected by aging), CD4^+^ T-helper cells, Tregs, and B-cells.

**Figure 6 cancers-13-01400-f006:**
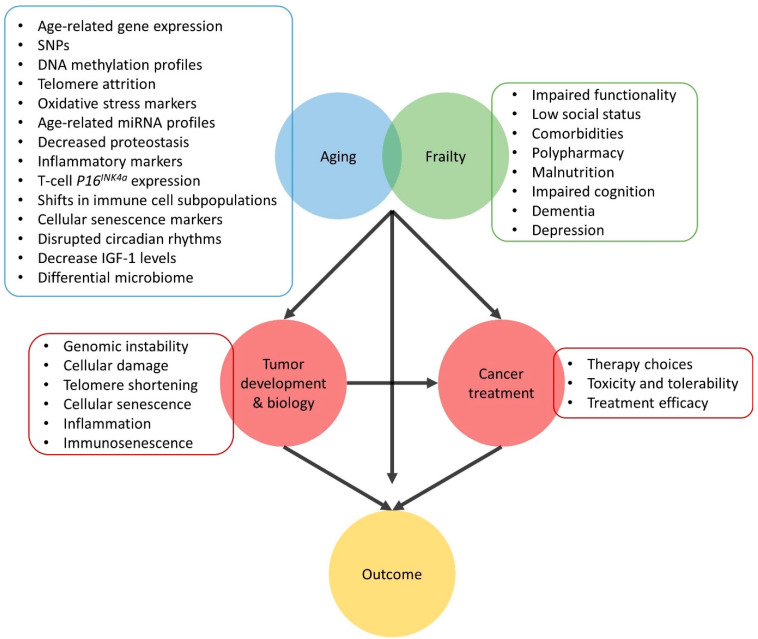
Schematic overview of strong interconnection between aging and cancer. The discussed biomarkers of aging are summarized as well as the clinical symptoms of frailty. The aging process and cancer share several biological processes impacting oncological decision making. However, treatment choices, toxicity and tolerability and treatment efficacy are highly influenced by age and more importantly frailty status. Taking all this together, age and frailty have an immense impact on the risk of developing cancer, cancer biology, prognosis, and therapy choices and thus the outcome for older patients with cancer.

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
