# Peer review of "Cancer and Aging: Two Tightly Interconnected Biological Processes"

_cancers, 2021, doi:10.3390/cancers13061400_

Round 1

Reviewer 1 Report

In this work, the authors  highlight the relevance of identifying biomarkers that might allow to understand the effective biological age of old cancer patients in order to give them an appropriate and personilized treatment. Overall the review is well written and clear. A comprehensive description of the interplay between aging and cancer has been made. The section of "Aging biomarkers" that should be a prognostic and predictive tool for aged cancer patients, should be implemented (ie circadian clocks, proteostasis, long-non coding RNA...) and better organized.

Minor points:

- Referencees should be cited next to the releative sentences along the Manuscript.

- In the caption 2.1 an extended description, specific references and a scheme of oxidative stress and DNA damage response are suggested.

-line 149: the authors should verify the data in the original paper regarding the percentage of p53 heterozygous and knockout mice.

-The work of Serrano's group, 2006 that exclude the p53 age promoting effect in telomere-driven aging must also be commented.

-line 155: the original paper Sharpless, 2001 should be inserted.

-Caption 2.3.2 is actually 2.3.3

-In this section authors should consider to give a description of the potential use of senolytic drugs in senescence-associated cancer (Campisi, 2019; Corina Amor, 2020; Mongiardi, 2021).

-Lines 393-399: It is not clear.

Author Response

The section of "Aging biomarkers" that should be a prognostic and predictive tool for aged cancer patients, should be implemented (ie circadian clocks, proteostasis, long-non coding RNA...) and better organized.

Thank you for your relevant comments and suggestions. The “Aging biomarker” section was reorganized and extra biomarkers were added. These additions are marked in the track changes. We acknowledge already in the introduction of chapter 3 that we only mention a selection of biomarkers for which most data exist.

Minor points:

  • References should be cited next to the releative sentences along the Manuscript.

Indeed, in some places, references were missing in important sentences. We have added or replaced various reference along the manuscript, these are all marked in red.

  • In the caption 2.1 an extended description, specific references and a scheme of oxidative stress and DNA damage response are suggested.

Thank you for this valuable suggestion, we have extended this section and added an overview of the DNA damage response after oxidative damage to make it more clear (Figure 1). 

  • Line 149: the authors should verify the data in the original paper regarding the percentage of p53 heterozygous and knockout mice.

We have checked the data in the original paper and could verify the numbers we reported.

  • The work of Serrano's group, 2006 that exclude the p53 age promoting effect in telomere-driven aging must also be commented.
  • Line 155: the original paper Sharpless, 2001 should be inserted.

The important works mentioned above were indeed missing and are now added to the manuscript.

  • Caption 2.3.2 is actually 2.3.3

We have corrected this typo.

  • Lines 393-399: It is not clear.

We acknowledge that this section was unclear, it was replaced and rewritten. It can now be find in the section “Aging biomarkers”, subtitle 3.15. Biomarker panels.

  • In this section authors should consider to give a description of the potential use of senolytic drugs in senescence-associated cancer (Campisi, 2019; Corina Amor, 2020 ; Mongiardi, 2021).

We are highly grateful for the comments/suggestions from both our reviewers listed above. We have added an extra part about the treatment of older patients with cancer, in which the mentioned points are being discussed in more detail.

We have also changed the title of our review to a better suiting one (marked in red).

Reviewer 2 Report

It is my pleasure to review this review on biology of cancer and aging.

Overall, it is well-written paper, comprehensively reviewing and summing the mechanistic connection between aging and cancer and aging biomarkers.

I only have the following minor comments and suggestions:

  • I would also suggest adding the microbiome as a potential aging or cancer biomarker.
  • Aging, diet or other lifestyle might also contribute to cancer development.
  • How about adding the relationship on aging and other cancer treatment, such as chemotherapy, radiotherapy, surgery? This study seems to mainly focus on immunotherapy.
  • I would recommend adding more epidemiology or human studies for each biomarkers, to make it more convincible.
  • Several figures could be combined to make it more comprehensive to understand.
  • A summary table might be helpful for readers to understand and digest.
  • Might also add more aging related symptoms, such as malnutrition, depression, dementia, and show how it connect aging and cancer development and survival.

Author Response

Thank you for your valuable suggestions.

  • I would also suggest adding the microbiome as a potential aging or cancer biomarker.

Thank you for this suggestion, the microbiome was added as a potential biomarker and is marked in the track changes.

  • I would recommend adding more epidemiology or human studies for each biomarkers, to make it more convincible.

For the biomarkers where epidemiology or human studies were missing, relevant information was added. This can also be seen in the track changes.

  • Several figures could be combined to make it more comprehensive to understand.

We agree and we have combined the figures concerning immunosenescence of the innate and adaptive immune system.

  • A summary table might be helpful for readers to understand and digest.

A schematic overview was added, discussing the different elements described in this review. It can be found at the end of the manuscript (Figure 6).

  • How about adding the relationship on aging and other cancer treatment, such as chemotherapy, radiotherapy, surgery? This study seems to mainly focus on immunotherapy.
  • Might also add more aging related symptoms, such as malnutrition, depression, dementia, and show how it connect aging and cancer development and survival.
  • Aging, diet or other lifestyle might also contribute to cancer development

We are highly grateful for the comments/suggestions from both our reviewers listed above. We have added an extra part about the treatment of older patients with cancer, in which the mentioned points are being discussed in more detail.

We have also changed the title of our review to a better suiting one (marked in red).